# Spatiotemporal Trends and Age–Period–Cohort Analysis for the Burden of Endometriosis-Related Infertility: An Analysis of the Global Burden of Disease Study 2019

**DOI:** 10.3390/jpm13091284

**Published:** 2023-08-22

**Authors:** Jingchun Liu, Wuyue Han, Haoyu Wang, Zhi Wang, Bingshu Li, Li Hong

**Affiliations:** Department of Obstetrics and Gynecology, Renmin Hospital of Wuhan University, Wuhan 430060, China2021283020221@whu.edu.cn (W.H.);

**Keywords:** endometriosis, infertility, global burden of disease, age–period–cohort model

## Abstract

Background: Endometriosis is a common nonfatal gynecological disease, and infertility is one of its main dangers. Endometriosis-related infertility causes serious damage to women’s health and places a burden on women of reproductive age. The aim of this study was to describe the current burden of endometriosis-associated infertility and to analyze its spatiotemporal trends. Methods: Age-standardized prevalence rate (ASPR) data from 1990 to 2019 for Endometriosis-related primary infertility (ERPI) and secondary infertility (ERSI) were obtained from the Global Burden of Disease Study (GBD) 2019. These data spanning three decades cover the global, sociodemographic index (SDI) regions, GBD regions, and 204 countries and territories. Spatiotemporal trends were analyzed by calculating the estimated annual percentage change (EAPC) and using a time–period–cohort model. Results: Globally, the ASPR of ERPI and ERSI showed a weak downward trend from 1990 to 2019, with EAPCs of −1.25 (95% CI: −1.39 to −1.11) and −0.6 (95% CI: −0.67 to −0.53), respectively. The spatiotemporal trends in ERPI and ERSI varied substantially between regions and age groups. When endometriosis-related infertility burden was linked to SDI values, a strong negative correlation was observed between the ASPR of ERSI and its EAPC and SDI values. When modeling with age–period–cohort, ERPI burden was found to be highest at ages 20–25 years, while ERSI burden was persistently higher at ages 20–45 years. Using 2000–2004 as the reference period, both ERPI and ERSI burden decreased with each year among women. Significant variability in burden between regions was found for the birth cohort factor. Conclusions: The global burden of endometriosis-related infertility declined minimally from 1990 to 2019. However, this burden varied considerably across regions, age groups, periods, and birth cohorts. The results of this study reflect spatiotemporal trends in the burden of endometriosis-related infertility over the study period and may be used to help improve health management, develop timely and effective prevention and control strategies, and provide epidemiologic theoretical evidence for reducing the burden for endometriosis-related infertility.

## 1. Introduction

Endometriosis is a common, nonfatal chronic disease manifested by the presence of endometrial-like tissue outside the uterus that has an insidious onset, inadequate detection of the disease, and a lack of noninvasive diagnostic methods, leading to delays and often misdiagnosis, delaying the patient’s access to optimal treatment [1,2]. Endometriosis poses many problems and impairments for women. Infertility is one of the major impairments of endometriosis, and about 50% of women suffering from infertility have a link to endometriosis [3]. Although the exact mechanism linking infertility to endometriosis is currently unknown, there is growing evidence of an association [4,5]. A growing number of studies are exploring the relationship between endometriosis and related infertility with the intention of finding appropriate therapeutic targets. However, the burden of infertility associated with endometriosis has not been reported in detail.

The Global Burden of Disease Study (GBD) 2019 is an international comprehensive statistical measure of disease and the associated impairments. The last brief assessment of the global burden of endometriosis was based on GBD 2017 [6]. In the current study, we performed an updated analysis of the burden of endometriosis-related infertility using the most recent data from GBD 2019, as demonstrated by the age-standardized prevalence rate (ASPR) at the global, regional, and national levels. In addition, we focused on age and social stratification in conjunction with age–period–cohort models in order to gain insight into the burden of infertility associated with endometriosis. This study provides statistical data on the temporal and spatial burden of infertility trends associated with endometriosis using the most recent comprehensive public health data. The aim is to better measure the current burden of prevalence of endometriosis-related infertility. The manifestation of these burdening advances may provide the basis for increasing women’s health awareness, raising awareness of endometriosis-associated infertility, and calling for early diagnosis and treatment of the primary disease.

## 2. Materials and Methods

### 2.1. Data Sources

The GBD 2019 database (https://ghdx.healthdata.org/gbd-2019, accessed on 10 February 2023) covers a wide range of population health and demographic data from across the globe obtained from census data, surveys, registries, indicators and estimates, administrative health data, and health-related financial data. Disease and impairment data and 95% uncertainty interval (UI) from 1990 to 2019 were obtained from the GBD 2019 public database (accessed on 10 February 2023). Our analysis focused on the burden of endometriosis-related infertility, including endometriosis-related primary infertility (ERPI) and endometriosis-related secondary infertility (ERSI). ASPR data were used to evaluate the burden for endometriosis-related infertility. For analysis of age-specific data on the burden of endometriosis, health data from the GBD 2019 were used for women aged 15 to 54 years, grouped by five years each. The GBD 2019 provided infertility data for women aged 15–49 years, also grouped by the five years, to assess the age-specific burden of endometriosis-associated infertility. Data for each region were provided by GBD 2019. Based on the SDI values, the 204 countries were divided into five regions with different SDI levels.

### 2.2. Data Definition

In GBD 2019, endometriosis was defined as N80-N80.9 to match ICD10. Specifically, cases confirmed by laparoscopic or pathologically confirmed pelvic examination were permitted [7]. In GBD 2019, primary infertility is defined as being present in couples who have not had a live birth, and want children and have been in a relationship for more than 5 years without using contraceptives. Secondary infertility was defined as being present in a couple who wanted a child and had been in a relationship for more than 5 years without contraceptive use since their last live birth [8]. Our study focused on a population of women with endometriosis and characterized by infertility identified through health surveys in GBD 2019.

The sociodemographic index (SDI) is a comprehensive assessment of national and regional per capita income, years of schooling, and fertility rates for women under 25 years of age, and allows for the classification of all countries and regions into five classes.

In our study, we first restricted the disease to endometriosis and then focused on infertility as an impairment to obtain ERPI and ERSI data. Consequently, women aged 15–49 years with endometriosis-related infertility were successfully acquired.

### 2.3. Calculation of the Estimated Annual Percentage Change

ASPR from 1990 to 2019 were used to assess burden of endometriosis-related infertility. Temporal trends in burden over three decades are captured by the estimated annual percentage change (EAPC). EAPC and its 95% confidence interval (CI) are obtained from the formula EAPC = 100 ∗ (exp (β) − 1), where β is the annual change in ln (ASPR). A positive value of EAPC and its 95% CI represents an upward trend; if both values are negative, it is a downward trend; otherwise, we consider it relatively stable.

The age–period–cohort model is based on a Poisson distribution and was used to assess independent estimates of the effects of age, period, and birth cohort on the burden of endometriosis-related infertility globally and in five SDI-level regions.

### 2.4. Construction of the Age–Period–Cohort Model

The age–period–cohort model is based on a Poisson distribution and was used to assess independent estimates of the effects of age, period, and birth cohort on the burden of endometriosis globally and in five SDI-level regions. The R packages “magrittr” and “dplyr” are used to construct the model. Here, the age factor is determined by the loading of aging factors. Period factors refer to changes in the load of anthropogenic factors in a given epoch. The variation of the cohort factor is attributed to the different exposure conditions of the population in different birth periods. In this case, the age, period, and birth cohort are each 5 years apart. The net drift represents the overall time trend, and a *p*-value of no more than 0.05 indicates a statistical difference. The longitudinal age curve is used to assess changes in disease burden due to age effects. Period rate ratios (RR) and cohort RR show period effects and birth cohort effects, with RR greater than 1 representing a higher relative risk of disease compared with the reference cohort and RR less than 1 representing a lower risk.

### 2.5. Statistical Analysis

Correlations between statistics, SDI, and EAPC were determined using Pearson correlation. The age–period–cohort model was evaluated by the Wald test. All data could be obtained free of charge from GBD 2019 except those specifically mentioned. All statistical analyzes and plots were generated using GraphPad Prism8 or R Studio (4.1.0).

## 3. Results

### 3.1. Region-Based Description of the Burden of Endometriosis-Related Infertility

Infertility is considered the main impairment posed by endometriosis. We analyzed the prevalence of primary and secondary endometriosis-related infertility (Appendix A and Figure 1). In 1990, the ASPR for endometriosis-associated primary infertility (ERPI) was 11.4 (95% UI: 5.52 to 19.88) and for endometriosis-associated secondary infertility (ERSI) was 30.49 (95% UI: 18.32 to 47.66). For 2019, the ASPR for ERPI is 8.45 (95% UI: 3.69 to 15.45) and the ERSI is 25.35 (95% UI: 15.31 to 39.91). Overall, the global burden of endometriosis-related infertility showed a decreasing trend over the 30-year period. Specifically, the EAPC for the burden of ERPI was −1.25 (95% CI: −1.39 to −1.11) and for ERSI was −0.6 (95% CI: −0.67 to −0.53).

At the SDI region level, for ERPI, ASPR was consistently lowest in the High-middle SDI region and consistently highest in the Low SDI region over the 30-year period. For ERSI, ASPR is lowest in the High SDI region and, as expected, consistently highest in the Low SDI region from 1990 to 2019. The burden of endometriosis-related infertility trended downward in all SDI regions, including ERPI and ERSI. Over the three decades, the burden has declined fastest in the Low-middle SDI region, with EAPCs in ASPR for total endometriosis-related infertility, ERPI, and ERSI of −1.38 (95% CI: −1.43 to −1.33), −2.18 (95% CI: −2.45 to −1.91), and −1.07 (95% CI: −1.19 to −0.96), respectively.

At the GBD level, the highest ASPR for ERPI was attributed to North Africa and the Middle East in both 1990 and 2019. In 1990, the ASPR for ERSI peaked in Oceania (ASPR = 39.55, 95% UI: 23.25 to 63.23), whereas in 2019 it trended upward in North Africa and the Middle East (ASPR = 59.87, 95% UI: 36.98 to 91.84). Over the 30-year period, the ASPR of ERPI tended to increase in Eastern Europe and the ASPR of ERSI increase in Eastern Europe and Andean Latin America, remained essentially flat in Western Europe, and was lower in all other regions.

Of the 204 countries and territories with endometriosis-related infertility, both primary and secondary, Yemen provided the highest ASPR in 1990. Regarding ERPI, Yemen had the highest ASPR in 1990, while Iran (Islamic Republic of) had the highest in 2019. Over the three decades, the burden of ERPI declined in a total of 186 of 204 countries, with Liberia showing the most significant decline in burden with an EAPC = −3.85 (95% CI: −4.38 to −3.33). A total of 12 countries showed an increasing trend in the ASPR of ERPI, with Sweden showing the highest value with EAPC = 2.02 (95% CI: 1.57 to 2.48). The ASPR was stable in the other countries. For ERSI, Taiwan (Province of China) had the highest ASPR in both 1990 and 2019. A total of 162 countries showed a decreasing ASPR of ERSI from 1990 to 2019, with Pakistan showing the most significant decrease in ASPR with EAPC = −3.43 (95% CI: −3.74 to −3.12). Nineteen countries showed an increased ASPR, led by Peru with EAPC = 1.86 (95% CI: 0.95 to 2.79).

### 3.2. Age-Based Description of the Burden of Endometriosis-Related Infertility

GBD 2019 counted women aged 15 to 49 years with endometriosis-related infertility. We analyzed the age-specific burden for ASPR of ERPI and ERSI according to age groups, respectively (Figure 2). The highest ASPR for ERPI occurred in the 20–24 age group, while ERSI was found in the 25–29 age group. Over the past three decades, the burden of ERPI and ERSI in different age groups has decreased worldwide and in all regions. However, in the low SDI region, the burden is consistently highest in the 15–44-year-olds and shifts to the middle SDI region among the 45–49 age group.

### 3.3. Impact of Sociodemographic Transition on the Burden of Endometriosis-Related Infertility

We were also made aware of the association between sociodemographic changes and the prevalence of endometriosis-related infertility (Figure 3A). For ERPI, there was a slight overall decrease in ASPR with increasing SDI. However, for ERSI, a significant downward trend was observed. This reflects the fact that the burden of disease in ERSI declined significantly with the development of SDI. We further examined the pattern of EAPC with SDI in ASPR for endometriosis-associated infertility (Figure 3B). For ERSI, the change in EAPC with SDI was relatively stable. Although this represents a relatively stable trend of a slow reduction in burden of ERSI, the regions with higher SDI still have EAPCs converging toward 0, indicating a potential risk of burden increase. In ERPI, the EAPC reaches its nadir at SDI values of 0.5–0.6, which represents the most significant reduction in burden of ERPI. Thereafter, it increases rapidly and reaches its highest point, approaching 0 at an SDI value of approximately 0.8. The EAPC of ASPR for ERSI appears to decrease further as the SDI value continues to increase.

### 3.4. Age–Period–Cohort Effect of Endometriosis-Related Infertility Burden

Our objective was to further understand the change in prevalence burden of ERPI and ERSI using the age–period–cohort model. The overall risk of impairment during the study period was lower in the ERPI population with a net drift = −1.22 (95% CI: −1.37 to −1.06) (Appendix A). The age effect of ERPI showed an inverted V-shaped trend (Figure 4). That is, the burden of ERPI peaks at the age of 20–24 years and decreases after the age of 25 years. The period RR showed a decreasing trend over time, indicating a relatively slow increase in risk until 2002 and a decreasing trend thereafter. The cohort RR showed a significant downward trend, reaching 1 in 1970. The trends between age, period, and cohort were similar to the overall trend at the global level for all five SDIs. However, in regions with high SDI values, the age interval between 25 and 29 years seems to be the one with the highest prevalence. It is also worth noting that, using 1970 as the reference birth year, the burden of ERPI declined more rapidly in women born after 1975 in areas with lower SDI values; although, birth cohort factors showed similar trends.

For ERSI, an overall decreasing trend in prevalence risk was observed at the global level, with a net drift of −0.84 (95% CI: −1.01 to −0.67), which was statistically significant. As for the age factor, prevalence tended to increase with age and then decreased, peaking at 25–29 years. Regarding SDI regions, although they all showed a general trend of first increasing and then decreasing with age, the vast majority of regions had the highest prevalence risk at age 25–29 years. However, the areas with high SDI peaked at age 40–44 years, showing a significant delay in the age peak of prevalence risk. Regarding period effects, the period RR tends to decrease over time, with 2000–2004 as the reference period. For the birth cohort with 1970 as the reference year, greater variability is seen at the global level. At the global level, the rate of increase in ERSI burden slows with increasing birth year for women born before 1970 and decreases each year for women born between 1970 and 1982, whereas the burden of ERSI increases again for women born in subsequent years. The birth cohort effect of ERSI shows a remarkably high degree of variability within SDI regions. In the high SDI region, results show that ERSI burden decreases with birth year for both women born before 1950 and women born after 1970; whereas, the opposite is true for women born between 1950 and 1970. In other SDI regions, the most consistent observation is that ERSI burden increases with birth year, a phenomenon that began to reverse in the 1970 birth cohort.

## 4. Discussion

Endometriosis is now considered a systemic disease and is not limited to the pelvis [1]. It can affect systemic metabolism, disrupt the immune system, cause inflammation, and lead to systemic pain [9,10,11,12]. In addition, endometriosis leads to an increased risk of future gynecologic diseases, both benign and malignant [13,14,15]. Infertility is one of the main impairments of endometriosis and can be associated with mechanical factors, ovarian factors, uterine factors, and pain [4,16]. Recent studies have shown that dysregulation of cytokines such as TNF-α, IFN-γ, and IL-97 associated with endometriosis is also an accomplice to infertility [17]. Furthermore, ferroptosis will increase the risk of infertility associated with endometriosis [18]. Endometriosis-related infertility affects many women of reproductive age worldwide. Early surgical treatment was once considered a feasible way to avoid progression to infertility. However, studies have reported that conception rates do not appear to improve significantly even after prior therapeutic surgery related to endometriosis, not to mention the irreversible damage that such surgery may cause to the women themselves [19]. Assisted reproductive technology is a well-established treatment with efficacy and success rates varying among individuals [20]. It has been reported that the use of synthetic progestins prior to in vitro fertilization and embryo transfer has no significant benefit on clinical outcomes in endometriosis-associated infertility [21]. A recent study suggests that astaxanthin, with its anti-inflammatory and antioxidant properties, may be a reliable intervention for the prevention of endometriosis-related infertility [22]. Endometriosis-associated infertility already causes physical and psychological distress and suffering for both patients and their partners [23,24], and understanding its long-term disease burden will help to increase the importance of preventing endometriosis from the beginning and managing endometriosis-associated infertility. Previous studies have reported an increasing prevalence of female infertility in recent years, but reports of endometriosis-related infertility are lacking [25]. In our study, we pooled global, multi-age data on endometriosis-related infertility over three decades and plotted trends to provide a multidimensional analysis of spatiotemporal burden. This study aims to better understand the current burden of endometriosis-related infertility.

In our study, the prevalence burden of endometriosis-related infertility showed a weak downward trend over three decades at the global level and in all SDI regions. Notably, the overall trend for ERPI and ERSI showed the same trend. However, the different SDI regions showed a split in prevalence burden, i.e., the burden was higher in the lower SDI regions and the prevalence burden was lower in the higher SDI regions. This differential status was particularly evident for ERSI, suggesting regional differences. Nevertheless, in the regions with lower SDI, the decline in burden was most rapid, indicating efforts in controlling endometriosis-related infertility. However, this rapid decline in disease burden may have a false-positive element. In short, our study spans three decades. During the past three decades, as medical care has advanced, so have the criteria for diagnosing endometriosis. This is what compels us to wonder whether overdiagnosis due to low-quality ultrasound technology may have occurred thirty years ago. In addition, we found that the prevalence burden of ERPI was hardly negatively correlated with SDI, except for SDI values of 0.3–0.5, which reflected a relatively high burden, and a slight lower burden for SDI values above 0.8. In contrast, the ASPR for ERSI showed a sensitive decline with increasing SDI values. This suggests that ERSI may be the major decreasing burden when SDI increases with social development and SDI. A favorable social environment may contribute to more comprehensive management and prevention of women’s health, including stricter control of the ERSI burden. In addition, the emergence of this phenomenon may be related to the quality of care. Countries and regions with rapid social development are often matched with better health care systems, such as assisted reproductive technology or laparoscopic techniques, which make it possible for infertility not to become a major problem. However, in poorer countries and regions, it can still be a huge problem. We also observed a negative sinusoidal trend in the EAPC of ASPR with SDI values in ERPI, which reflected a reversion in the EAPC in the range of SDI values 0.75–0.85, suggesting a potential risk of burden rebound. This change was not seen in secondary infertility.

The prevalence of endometriosis-associated infertility varies by age group. The highest prevalence of ERPI is between 20 and 24 years of age, with the burden decreasing with age. Regional differences are also seen in younger women, with higher prevalence in areas with lower SDI. These differences become negligible with increasing age. In ERSI, all women aged 20 to 44 years had a high prevalence burden, and regional differences continued to exist, manifested by a negative correlation between SDI level and burden. The age–period–cohort model showed a decline in the risk of endometriosis-related infertility from 1990 to 2019, with a rebound risk for ERSI prevalence in high SDI areas in recent years. The age effect had a greater impact on infertility risk for ERPI and was most pronounced among younger women. For ERSI, prevalence risk was consistently higher among young and middle-aged women, particularly in the high SDI region that had a delayed age peak in prevalence. This suggests that the potentially advanced aging burden of endometriosis-related infertility may be a major problem in the future, for which we need to start addressing. For the cohort effect, the early birth cohort had a relatively high risk of infertility. However, for ERSI, women born after 1995 also suffered a high risk, as was most evident in the low–middle SDI regions. These data remind us that prevention and control of endometriosis-related infertility is an ongoing struggle, requiring consideration of the impact of social context and population age.

Overall, these results suggest that global efforts to combat endometriosis-related infertility have been reasonably successful. However, the age-related and spatially uneven distribution of the endometriosis-related infertility burden may undermine this progress. Although higher SDI areas currently appear to reflect a lower burden, there is a risk that the burden will rise again if efforts remain inadequate. In addition, efforts to curb the burden of endometriosis-related infertility must be aligned with actual health needs and social environments, taking into account disease background, regional culture, and age composition to be effective in reducing the burden of disease. The GBD 2019 study takes into account influencing factors such as social context and population age on the changing global burden of disease landscape to inform disease and impairment prevention and control strategies.

Several limitations provide room for further development of this study. First, data on endometriosis and associated infertility in some countries and regions need further refinement to provide a more accurate and reasonable estimate of disease burden. The burden of disease and harms do not currently cover the entire age range, which will lead to the absence of a significant amount of important data. Because our study spanned three decades, the criteria for diagnosis and confirmation of disease have evolved over those three decades, which may lead to inevitable over- or underdiagnosis. We had to account for some of the resulting false-positive or false-negative results. Second, infertility data are controversial. This is because in some surveys only married women were interviewed, but the final prevalence estimates suggest infertility among couples. In such cases, it is unrealistic to determine which partner is the cause of infertility as soon as possible. At the same time, there is no complete guarantee at the time of the survey that infertility is caused by endometriosis. These two limitations may have led to a different view from other studies available to date. Third, this study lacks data on factors that influence endometriosis, such as environment, diet, lifestyle, or metabolism. These additional data would help to address the burden of endometriosis. Medical technologies such as laparoscopy and advances in ultrasound diagnostics may contribute to the diagnosis and treatment of endometriosis-related infertility. This is a direction that could be explored in depth in the future. Finally, this study only runs through 2019 and does not consider the potential impact of COVID-19 on endometriosis and related infertility. This could have a significant impact on the landscape for global disease and impairment, as lack of health monitoring, inadequate diagnosis, and limited treatment options could reduce global efforts to combat endometriosis and related infertility.

## 5. Conclusions

In summary, the GBD 2019 study provides a comprehensive, high-quality estimate of the burden of disease for endometriosis-related infertility. Over the past three decades, the burden of endometriosis-related infertility has declined more or less significantly in a concerted worldwide effort. However, global estimates of the burden of endometriosis-related infertility vary widely, highlighting disparities in the global burden, particularly in less developed regions. In addition, age differences show disparities in disease burden, and the potential risk of disease aging must be taken seriously. Public health leaders should develop more complex responses based on specific social contexts and age distribution characteristics, actively promote women’s health knowledge, call for attention to endometriosis, and improve medical care in order to fundamentally reduce the burden of endometriosis-related infertility and regional disparities to advance women’s health.

## Figures and Tables

**Figure 1 jpm-13-01284-f001:**
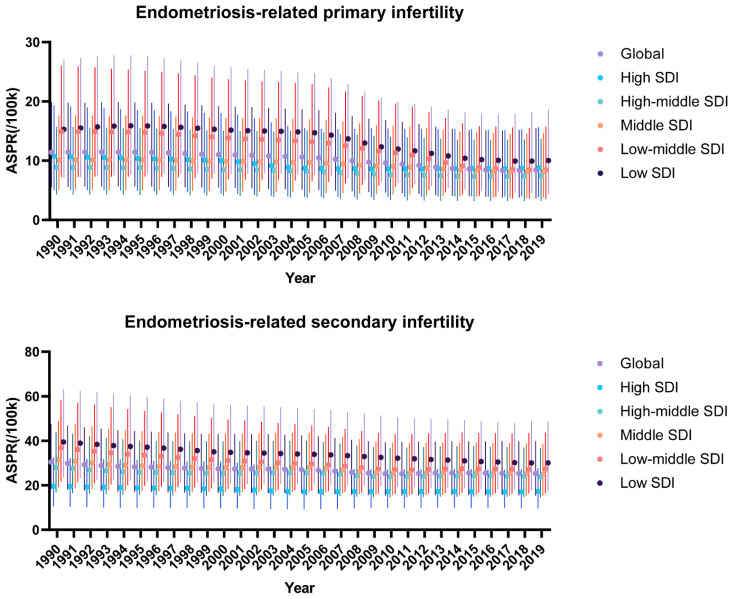
Spatiotemporal trends in ASPR for ERPI and ERSI over three decades. The median ASPR for ERPI and ERSI and its 95% UI are shown for the global and SDI regional levels. UI: uncertainty interval.

**Figure 2 jpm-13-01284-f002:**
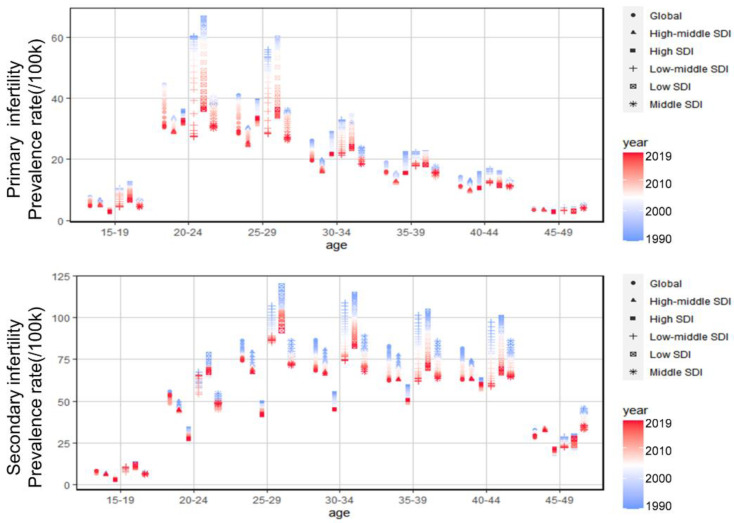
Age-specific prevalence rate of ERPI and ERSI over three decades.

**Figure 3 jpm-13-01284-f003:**
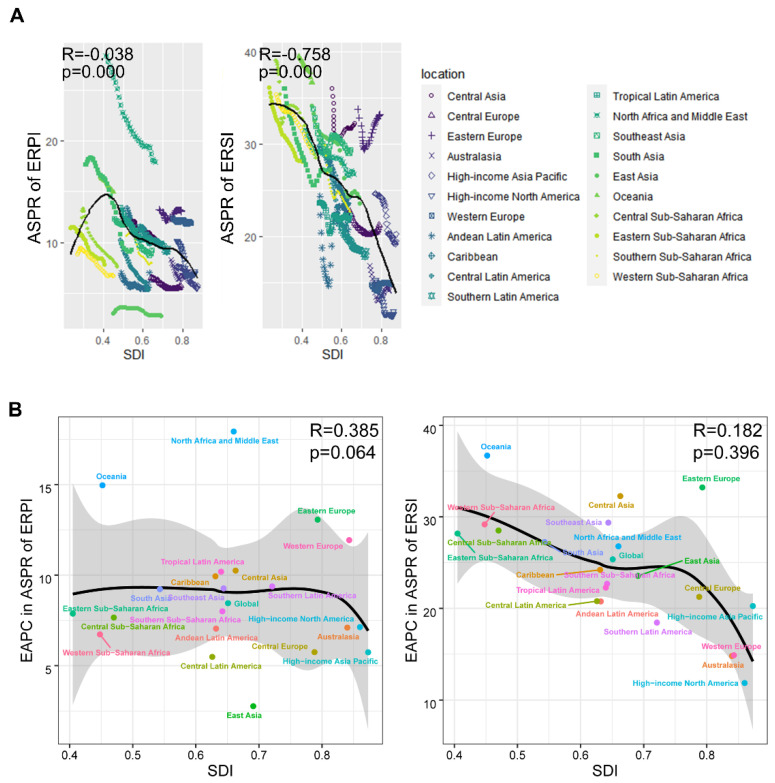
Impact of sociodemographic transition on the burden for ASPR of endometriosis-related infertility. (**A**) Association of SDI with ASPR in endometriosis-associated infertility. The same location markers represent changes in the same location over a 30-year period. (**B**) Association of SDI with EAPC in ASPR of endometriosis-associated infertility.

**Figure 4 jpm-13-01284-f004:**
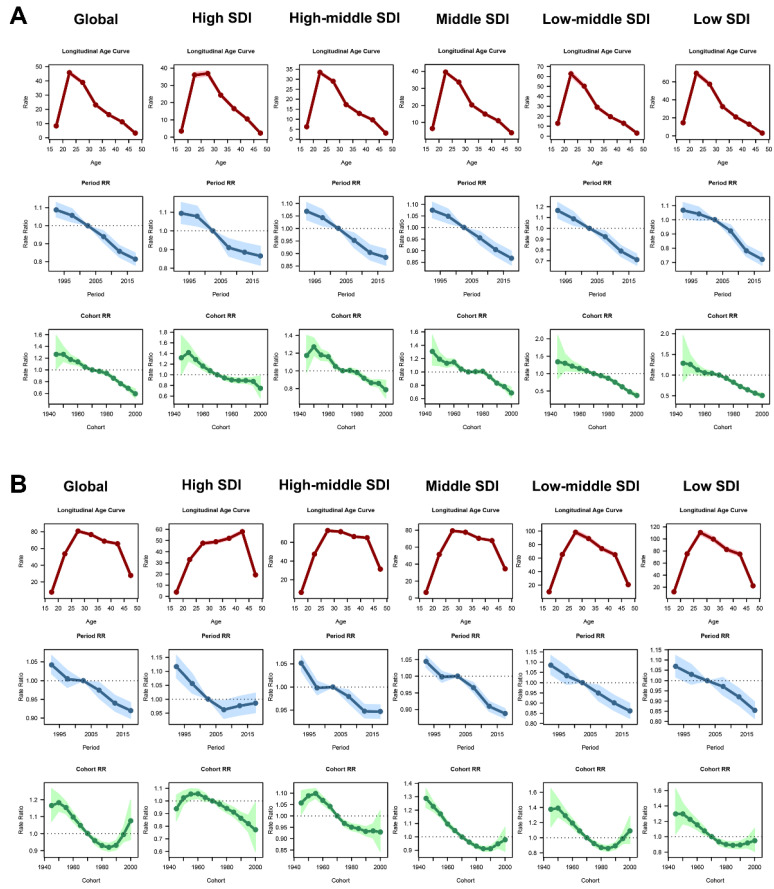
Age–period–cohort effect of burden for ASPR of endometriosis-related infertility. (**A**) Age–period–cohort model of ERPI. (**B**) Age–period–cohort model of ERSI. Red reflects the age factor, blue is the period factor, and green is the birth cohort factor.

## Data Availability

All data are openly and freely available from GBD 2019 (https://vizhub.healthdata.org/gbd-results/, accessed on 10 February 2023). Relevant codes can be obtained by contacting the corresponding author.

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
