# Peer review of "Spatiotemporal Trends and Age–Period–Cohort Analysis for the Burden of Endometriosis-Related Infertility: An Analysis of the Global Burden of Disease Study 2019"

_jpm, 2023, doi:10.3390/jpm13091284_

Round 1

Reviewer 1 Report

The manuscript ‘ Spatiotemporal Trends and Age-Period-Cohort Analysis for the 2 Burden of Endometriosis-Related Infertility: A Systematic 3 Analysis of the Global Burden of Disease Study 2019’ is well written en performed and deserves publication.

Therefore consider the comments below as suggestions to improve the manuscript.

-        Title : the word systematic is somewhat confusing with systematic review

Abstract. I had to read 3 times. My impression is that less detail of methods (such omitting as l17 estimated annual percentage change (EAPC) does not add to the message. ‘

-         with a minimum value of 0.55 and a maximum value of 26 0.8; ,’ is not clear. The conclusion could be more informative eg are changes related to the socioeconomic situation and the availability of laparoscopy  

Introduction: parts of the second paragraph belong to materials and methods.

M&M

-        What is the content of the GBD 2019 public database: hospital based discharge diagnosis?

-        Stress that durations of 5 years for primary and secondary infertility are not standard

-        Line 91: ln( age minus standardised rate) : I do not understand

-        Statistics : which package

Results

-        Fig 1 The suggestion is to delete the blobal and the total infertility. Since not adding to the conclusions

-        Fig 1: I am not a statistcian and EAPC is not very clear to me. -However, I would be surprised that a multivariate analysis of year AND SDI would not show a decrease in countries with a lower SDI. As a clinician, I would think of IVF in the richer countries and accessibility and quality of laparoscopy in the poorer countries. Results are nice but hard to read.

-        Age based description of the burden. : Maybe I do not understand - I am missing the prevalence of endometriosis which should decrease over time if I understand

- fig 3 is even more difficult to understand

In conclusion, this is a nice and well-performed paper, which, unfortunately, is very difficult to understand. Also, the figures, although nice are difficult to grasp

The authors are strongly suggested to revise the paper, trying to make the text understandable for the median reader and the figures more stand-alone. It will be a pleasure to review the discussion accordingly.

In materials and methods, it should be clear what the content is: If I understand well, these are women with infertility that were diagnosed with endometriosis, or is it vice versa . 

This nice paper needs revision for the reader to understand and for the authors to generate references

Reviewer 2 Report

The relationship between endometriosis and infertility is one of the most complex possible. For this reason study such as this must be taken with extremely caution when drawing conclusions and suggesting future policy . Given that , it's interesting to note that overall the situation has not worsen and probably slightly improved .  

Only minor revision is needed

Author Response

Thank you for your review. In the revised manuscript, we made extensive edits and changes to the language. We hope you are satisfied with the new manuscript.

Reviewer 3 Report

The manuscript "Spatiotemporal Trends and Age-Period-Cohort Analysis for the Burden of Endometriosis-Related Infertility: A Systematic Analysis of the Global Burden of Disease Study 2019 " is an interesting study on the correlation between endometriosis and infertility in the last 30 years. The work is complete and well structured, giving important data to scientific literature. The design of the project is appropriate and the results are significant. The statistical analysis is well conducted and the language is acceptable. The authors declare that "the aim is to use the data to promote women's health awareness, raise awareness of endometriosis-related infertility, and call for early diagnosis and treatment of the original disease": I think that it is not the aim of the study, but a potential future effect of studies like this one, therefore I suggest to modify this sentence. The main concern about this work is the diagnosis: considering the improvement in the last 30 years of the diagnostic tools and scientific knowledge of clinicians about endometriosis, a diagnosis made in 2018 could be not equal to a diagnosis in 1991; in 1991 an overdiagnosis of endometriosis could be related to low quality of ultrasound. It could be a bias of the work and the authors have to better underline this aspect as a limitation of the study. What are the actual clinical implications of this study? it is important to report the results obtained by the authors in the context of clinical practice and to adequately highlight what contribution this study adds to the literature already existing on the topic and to future study perspectives. Otherwise, it represents a valid work and it gives the opportunity to focus attention on possible future knowledge in Endometriosis and infertility.

Author Response

We sincerely thank all reviewers for their insightful comments. Based on these comments, we revised the manuscript extensively for content, language, and readability. Similarly, the figures and tables have been revised accordingly. In this revised manuscript, we have marked the locations of the changes in red.

Additionally, in the responses to reviewers, reviewers' comments are marked in black and responses are marked in red.

Thanks again to all the reviewers for their efforts. We hope that the new revised manuscript is to your satisfaction.

Point-by-point response

Reviewer 3

The manuscript "Spatiotemporal Trends and Age-Period-Cohort Analysis for the Burden of Endometriosis-Related Infertility: A Systematic Analysis of the Global Burden of Disease Study 2019 " is an interesting study on the correlation between endometriosis and infertility in the last 30 years. The work is complete and well structured, giving important data to scientific literature. The design of the project is appropriate and the results are significant. The statistical analysis is well conducted and the language is acceptable. 

Thank you for your careful review. We hope that our study will provide a theoretical basis for an in-depth study of endometriosis-associated infertility. And we hope that this study will evoke a deeper awareness of women's health.

The authors declare that "the aim is to use the data to promote women's health awareness, raise awareness of endometriosis-related infertility, and call for early diagnosis and treatment of the original disease": I think that it is not the aim of the study, but a potential future effect of studies like this one, therefore I suggest to modify this sentence.

Thank you for your review, we agree with you. Therefore, we have changed this part of the description in the revised manuscript. We emphasize that the aim of this purpose of this study is to better measure the current burden of endometriosis-associated infertility. And, it is our hope that these burdensome advances may provide the basis for increased awareness of women's health, greater recognition of endometriosis-related infertility, and a call for early diagnosis and treatment of the primary disease.

Thanks again for your careful review.

The main concern about this work is the diagnosis: considering the improvement in the last 30 years of the diagnostic tools and scientific knowledge of clinicians about endometriosis, a diagnosis made in 2018 could be not equal to a diagnosis in 1991; in 1991 an overdiagnosis of endometriosis could be related to low quality of ultrasound. It could be a bias of the work and the authors have to better underline this aspect as a limitation of the study.

Thank you for your careful review and constructive comments. We couldn't agree with you more. Due to the long period of time over which the study was designed, medical opinions and techniques have inevitably evolved, and the diagnosis of disease has changed, which will inevitably lead to false-positive or false-negative results. We must mention this in the study. Therefore, we have supplemented this perspective in our discussion, including in the sections describing changes in the burden of disease and describing the limitations of the study. Thank you again for your valuable comments.

What are the actual clinical implications of this study? it is important to report the results obtained by the authors in the context of clinical practice and to adequately highlight what contribution this study adds to the literature already existing on the topic and to future study perspectives. Otherwise, it represents a valid work and it gives the opportunity to focus attention on possible future knowledge in Endometriosis and infertility.

Thank you for your careful review. We agree that research should be conducted based on clinical realities. Our study was a public health and epidemiological analysis based on big data, and no clinical data were collected and analyzed from current centers. For this reason, our study represents an epidemiologic strategy. As we mentioned in the manuscript, we hope that this study will provide a theoretical basis for the epidemiologic prevention and control of endometriosis-associated infertility. In addition, we have supplemented the discussion section with some current clinical studies related to endometriosis. Thanks again for your insightful review.

Round 2

Reviewer 1 Report

The manuscript has improved but remains very difficult to read and understand for non epidemiologists.

The images remain difficult to understand e.g.

-Fig 1 Unclear whether SD or confidence limits are given

-Fig 2 The legend indication high and low SDI does not seem to correspond to this graph

 -Fig 3 A : I think the regions are represented, but even after enlarging the image I cannot identify regions in the graph

-Fig 4 : legends are unreadable 

Author Response

Thank you for your review. We have revised and responded to these comments.

(1) Thank you for your comments. We apologize for not writing clearly about the meaning of Figure 1. In the revised manuscript, we have supplemented it by pointing out that the markers are the median and 95% UI intervals of the ASPR.

(2) After our confirmation, there is no problem with the matching of Figure 2 and its legend. Since this figure is generated by R, the legends are arranged in initial alphabetical order, not global/high SDI/high-middle SDI/middle SDI/low-middle SDI/low SDI.

(3) Thank you for the correction. For Figure 3A, locations can be identified by shape and color. For Figure 3B, specific names have been labeled. We have confirmed that the clarity, size, and DPI of the image are up to standard and are clearly recognizable in our documentation.We consider that this may be related to the compression of the image quality by the document. Therefore, we uploaded the original figures in a zip archive. Thanks again for your review.

(4) Figure 4 seems to encounter a similar problem as Figure 3. Again, we examined it and after zooming in were able to clean and identify all the legends.In order to keep the size of the image aesthetically pleasing and in focus, we did not over-enlarge the font in this revision. The original images can also be found in the zip archive.

Thanks again for your review. Your review is vital to our manuscript!